# Nine best practices for research software registries and repositories

Daniel Garijo[1], Hervé Ménager[2], Lorraine Hwang[3], Ana Trisovic[4], Michael Hucka[5], Thomas Morrell[5], Alice Allen[6], Task Force on Best Practices for Software Registries[7], and SciCodes Consortium[8]

[1] Universidad Politécnica de Madrid, Madrid, Spain
[2] Institut Pasteur, Université Paris Cité, Bioinformatics and Biostatistics Hub, Paris, France
[3] University of California, Davis, Davis, California, United States
[4] Harvard University, Boston, Massachusetts, United States
[5] California Institute of Technology, Pasadena, California, United States
[6] University of Maryland, College Park, MD, United States
[7] FORCE11 Software Citation Implementation Working Group
[8] Consortium of Scientific Software Registries and Repositories

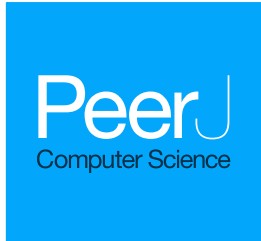

## ABSTRACT

Scientific software registries and repositories improve software findability and research transparency, provide information for software citations, and foster preservation of computational methods in a wide range of disciplines. Registries and repositories play a critical role by supporting research reproducibility and replicability, but developing them takes effort and few guidelines are available to help prospective creators of these resources. To address this need, the FORCE11 Software Citation Implementation Working Group convened a Task Force to distill the experiences of the managers of existing resources in setting expectations for all stakeholders. In this article, we describe the resultant best practices which include defining the scope, policies, and rules that govern individual registries and repositories, along with the background, examples, and collaborative work that went into their development. We believe that establishing specific policies such as those presented here will help other scientific software registries and repositories better serve their users and their disciplines.

## INTRODUCTION

Research software is an essential constituent in scientific investigations (*Wilson et al., 2014*; *Momcheva & Tollerud, 2015*; *Hettrick, 2018*; *Lamprecht et al., 2020*), as it is often used to transform and prepare data, perform novel analyses on data, automate manual processes, and visualize results reported in scientific publications (*Howison & Herbsleb, 2011*). Research software is thus crucial for reproducibility and has been recognized by the scientific community as a research product in its own right—one that should be properly described, accessible, and credited by others (*Smith, Katz & Niemeyer, 2016*; *Chue Hong et al., 2021*). As a result of the increasing importance of computational methods, communities such as Research Data Alliance (RDA) (*Berman & Crosas, 2020*) (https://www.rd-alliance.org/) and FORCE11 (*Bourne et al., 2012*) (https://www.force11.org/)

Corresponding author
Daniel Garijo, daniel.garijo@upm.es

emerged to enable collaboration and establish best practices. Numerous software services that enable open community development of and access to research source code, such as GitHub (https://github.com/) and GitLab (https://gitlab.com), appeared and found a role in science. General-purpose repositories, such as Zenodo (*CERN & OpenAIRE, 2013*) and FigShare (*Thelwall & Kousha, 2016*), have expanded their scope beyond data to include software, and new repositories, such as Software Heritage (*Di Cosmo & Zacchiroli, 2017*), have been developed specifically for software. A large number of domain-specific research software registries and repositories have emerged for different scientific disciplines to ensure dissemination and reuse among their communities (*Gentleman et al., 2004*; *Peckham, Hutton & Norris, 2013*; *Greuel & Sperber, 2014*; *Allen & Schmidt, 2015*; *Gil, Ratnakar & Garijo, 2015*; *Gil et al., 2016*).

*Research software registries* are typically indexes or catalogs of software metadata, without any code stored in them; while in *research software repositories*, software is both indexed *and* stored (*Lamprecht et al., 2020*). Both types of resource improve software discoverability and research transparency, provide information for software citations, and foster preservation of computational methods that might otherwise be lost over time, thereby supporting research reproducibility and replicability. Many provide or are integrated with other services, including indexing and archival services, that can be leveraged by librarians, digital archivists, journal editors and publishers, and researchers alike.

Transparency of the processes under which registries and repositories operate helps build trust with their user communities (*Yakel et al., 2013*; *Frank et al., 2017*). However, many domain research software resources have been developed independently, and thus policies amongst such resources are often heterogeneous and some may be omitted. Having specific policies in place ensures that users and administrators have reference documents to help define a shared understanding of the scope, practices, and rules that govern these resources.

Though recommendations and best practices for many aspects of science have been developed, no best practices existed that addressed the operations of software registries and repositories. To address this need, a Best Practices for Software Registries Task Force was proposed in June 2018 to the FORCE11 Software Citation Implementation Working Group (SCIWG) (https://github.com/force11/force11-sciwg). In seeking to improve the services software resources provide, software repository maintainers came together to learn from each other and promote interoperability. Both common practices and missing practices unfolded in these exchanges. These practices led to the development of nine best practices that set expectations for both users and maintainers of the resource by defining management of its contents and allowed usages as well as clarifying positions on sensitive issues such as attribution.

In this article, we expand on our pre-print "Nine Best Practices for Research Software Registries and Repositories: A Concise Guide" (*Task Force on Best Practices for Software Registries et al., 2020*) to describe our best practices and their development. Our guidelines are actionable, have a general purpose, and reflect the discussion of a community of more than 30 experts who handle over 14 resources (registries or

repositories) across different scientific domains. Each guideline provides a rationale, suggestions, and examples based on existing repositories or registries. To reduce repetition, we refer to registries and repositories collectively as "resources."

The remainder of the article is structured as follows. We first describe background and related efforts in "Background", followed by the methodology we used when structuring the discussion for creating the guidelines (Methodology). We then describe the nine best practices in "Best Practices for Repositories and Registries", followed by a discussion (Discussion). "Conclusions" concludes the article by summarizing current efforts to continue the adoption of the proposed practices. Those who contributed to the development of this article are listed in Appendix A, and links to example policies are given in Appendix B. Appendix C provides updated information about resources that have participated in crafting the best practices and an overview of their main attributes.

## BACKGROUND

In the last decade, much was written about a reproducibility crisis in science (*Baker, 2016*) stemming in large part from the lack of training in programming skills and the unavailability of computational resources used in publications (*Merali, 2010*; *Peng, 2011*; *Morin et al., 2012*). On these grounds, national and international governments have increased their interest in releasing artifacts of publicly-funded research to the public (*Office of Science & Technology Policy, 2016*; *Directorate-General for Research & Innovation (European Commission), 2018*; *Australian Research Council, 2018*; *Chen et al., 2019*; *Ministère de l'Enseignement supérieur, de la Recherche et de l'Innovation, 2021*), and scientists have appealed to colleagues in their field to release software to improve research transparency (*Weiner et al., 2009*; *Barnes, 2010*; *Ince, Hatton & Graham-Cumming, 2012*) and efficiency (*Grosbol & Tody, 2010*). Open Science initiatives such as RDA and FORCE11 have emerged as a response to these calls for greater transparency and reproducibility. Journals introduced policies encouraging (or even requiring) that data and software be openly available to others (*Editorial Staff, 2019*; *Fox et al., 2021*). New tools have been developed to facilitate depositing research data and software in a repository (*Baruch, 2007*; *CERN & OpenAIRE, 2013*; *Di Cosmo & Zacchiroli, 2017*; *Clyburne-Sherin, Fei & Green, 2019*; *Brinckman et al., 2019*; *Trisovic et al., 2020*) and consequently, make them citable so authors and other contributors gain recognition and credit for their work (*Soito & Hwang, 2017*; *Du et al., 2021*).

Support for disseminating research outputs has been proposed with FAIR and FAIR4RS principles that state shared digital artifacts, such as data and software, should be Findable, Accessible, Interoperable, and Reusable (*Wilkinson et al., 2016*; *Lamprecht et al., 2020*; *Katz, Gruenpeter & Honeyman, 2021*; *Chue Hong et al., 2021*). Conforming with the FAIR principles for published software (*Lamprecht et al., 2020*) requires facilitating its discoverability, preferably in domain-specific resources (*Jiménez et al., 2017*). These resources should contain machine-readable metadata to improve the discoverability (Findable) and accessibility (Accessible) of research software through search engines or from within the resource itself. Furthering interoperability in FAIR is aided through the adoption of community standards *e.g.*, schema.org (*Guha, Brickley & Macbeth, 2016*) or

the ability to translate from one resource to another. The CodeMeta initiative (*Jones et al., 2017*) achieves this translation by creating a "Rosetta Stone" which maps the metadata terms used by each resource to a common schema. The CodeMeta schema (https://codemeta.github.io/) is an extension of schema.org which adds ten new fields to represent software-specific metadata. To date, CodeMeta has been adopted for representing software metadata by many repositories (https://hal.inria.fr/hal-01897934v3/codemeta).

As the usage of computational methods continues to grow, recommendations for improving research software have been proposed (*Stodden et al., 2016*) in many areas of science and software, as can be seen by the series of "Ten Simple Rules" articles offered by PLOS (*Dashnow, Lonsdale & Bourne, 2014*), sites such as AstroBetter (https://www.astrobetter.com/), courses to improve skills such as those offered by The Carpentries (https://carpentries.org/), and attempts to measure the adoption of recognized best practices (*Serban et al., 2020*; *Trisovic et al., 2022*). Our quest for best practices complements these efforts by providing guides to the specific needs of research software registries and repositories.

## METHODOLOGY

The best practices presented in this article were developed by an international Task Force of the FORCE11 Software Citation Implementation Working Group (SCIWG). The Task Force was proposed in June 2018 by author Alice Allen, with the goal of developing a list of best practices for software registries and repositories. Working Group members and a broader group of managers of domain specific software resources formed the inaugural group. The resulting Task Force members were primarily managers and editors of resources from Europe, United States, and Australia. Due to the range in time zones, the Task Force held two meetings 7 h apart, with the expectation that, except for the meeting chair, participants would attend one of the two meetings. We generally refer to two meetings on the same day with the singular "meeting" in the discussions to follow.

The inaugural Task Force meeting (February, 2019) was attended by 18 people representing 14 different resources. Participants introduced themselves and provided some basic information about their resources, including repository name, starting year, number of records, and scope (discipline-specific or general purpose), as well as services provided by each resource (*e.g.*, support of software citation, software deposits, and DOI minting). Table 1 presents an overview of the collected responses, which highlight the efforts of the Task Force chairs to bring together both discipline-specific and general purpose resources. The "Other" category indicates that the answer needed clarifying text (*e.g.*, for the question "is the repository actively curated?" some repositories are not manually curated, but have validation checks). Appendix C provides additional information on the questions asked to resource managers (Table C.1) and their responses (Tables C.2–C.4).

During the inaugural Task Force meeting, the chair laid out the goal of the Task Force, and the group was invited to brainstorm to identify commonalities for building a list of best practices. Participants also shared challenges they had faced in running their resources

**Table 1 Overview of the information shared by the 14 resources which participated in the first Task Force meeting.**

| Question | #Yes | #No | #Other |
|---|---|---|---|
| Is the resource discipline-specific? | 6 | 8 | 0 |
| Does the resource accept software only? | 8 | 6 | 0 |
| Does the resource require a software deposit? | 2 | 12 | 0 |
| Does the resource accept software deposits? | 10 | 4 | 0 |
| Can the resource mint DOIs? | 6 | 8 | 0 |
| Is the resource actively curated? | 10 | 1 | 3 |
| Can the resource be used to cite software? | 11 | 2 | 1 |

and policies they had enacted to manage these resources. The result of the brainstorming and discussion was a list of ideas collected in a common document.

Starting in May 2019 and continuing through the rest of 2019, the Task Force met on the third Thursday of each month and followed an iterative process to discuss, add to, and group ideas; refine and clarify the ideas into different practices, and define the practices more precisely. It was clear from the onset that, though our resources have goals in common, they are also very diverse and would be best served by best practices that were descriptive rather than prescriptive. We reached consensus on whether a practice should be a *best* practice through discussion and informal voting. Each best practice was given a title and a list of questions or needs that it addressed.

Our initial plan aimed at holding two Task Force meetings on the same day each month, in order to follow a common agenda with independent discussions built upon the previous month's meeting. However, the later meeting was often advantaged by the earlier discussion. For instance, if the early meeting developed a list of examples for one of the guidelines, the late meeting then refined and added to the list. Hence, discussions were only duplicated when needed, *e.g.*, where there was no consensus in the early group, and often proceeded in different directions according to the group's expertise and interest. Though we had not anticipated this, we found that holding two meetings each month on the same day accelerated the work, as work done in the second meeting of the day generally continued rather than repeating work done in the first meeting.

The resulting consensus from the meetings produced a list of the most broadly applicable practices, which became the initial list of best practices participants drew from during a two-day workshop, funded by the Sloan Foundation and held at the University of Maryland College Park, in November, 2019 (Scientific Software Registry Collaboration Workshop). A goal of the workshop was to develop the final recommendations on best practices for repositories and registries to the FORCE11 SCIWG. The workshop included participants outside the Task Force resulting in a broader set of contributions to the final list. In 2020, this group made additional refinements to the best practices during virtual meetings and through online collaborative writing producing in the guidelines described in the next section. The Task Force then transitioned into the SciCodes consortium (http://scicodes.net). SciCodes is a permanent community for research

software registries and repositories with a particular focus on these best practices. SciCodes continued to collect information about involved registries and repositories, which are listed in Appendix C. We also include some analysis of the number of entries and date of creation of member resources. Appendix A lists the people who participated in these efforts.

# BEST PRACTICES FOR REPOSITORIES AND REGISTRIES

Our recommendations are provided as nine separate policies or statements, each presented below with an explanation as to why we recommend the practice, what the practice describes, and specific considerations to take into account. The last paragraph of each best practice includes one or two examples and a link to Appendix B, which contains many examples from different registries and repositories.

These nine best practices, though not an exhaustive list, are applicable to the varied resources represented in the Task Force, so are likely to be broadly applicable to other scientific software repositories and registries. We believe that adopting these practices will help document, guide, and preserve these resources, and put them in a stronger position to serve their disciplines, users, and communities[1].

## Provide a public scope statement

The landscape of research software is diverse and complex due to the overlap between scientific domains, the variety of technical properties and environments, and the additional considerations resulting from funding, authors' affiliation, or intellectual property. A scope statement clarifies the type of software contained in the repository or indexed in the registry. Precisely defining a scope, therefore, helps those users of the resource who are looking for software to better understand the results they obtained.

Moreover, given that many of these resources accept submission of software packages, providing a precise and accessible definition will help researchers determine whether they should register or deposit software, and curators by making clear what is out of scope for the resource. Overall, a public scope manages the expectations of the potential depositor as well as the software seeker. It informs both what the resource does and does not contain.

The scope statement should describe:

- What is accepted, and acceptable, based on criteria covering scientific discipline, technical characteristics, and administrative properties
- What is not accepted, *i.e.*, characteristics that preclude their incorporation in the resource
- Notable exceptions to these rules, if any

Particular criteria of relevance include the scientific community being served and the types of software listed in the registry or stored in the repository, such as source code, compiled executables, or software containers. The scope statement may also include criteria that must be satisfied by accepted software, such as whether certain software quality metrics must be fulfilled or whether a software project must be used in published

---

[1] Please note that the information provided in this article does not constitute legal advice.

research. Availability criteria can be considered, such as whether the code has to be publicly available, be in the public domain and/or have a license from a predefined set, or whether software registered in another registry or repository will be accepted.

An illustrating example of such a scope statement is the editorial policy (https://ascl.net/wordpress/submissions/editiorial-policy/) published by the Astrophysics Source Code Library (ASCL) (*Allen et al., 2013*), which states that it includes only software source code used in published astronomy and astrophysics research articles, and specifically excludes software available only as a binary or web service. Though the ASCL's focus is on research documented in peer-reviewed journals, its policy also explicitly states that it accepts source code used in successful theses. Other examples of scope statements can be found in Appendix B.

## Provide guidance for users

Users accessing a resource to search for entries and browse or retrieve the description(s) of one or more software entries have to understand how to perform such actions. Although this guideline potentially applies to many public online resources, especially research databases, the potential complexity of the stored metadata and the curation mechanisms can seriously impede the understandability and usage of software registries and repositories.

User guidance material may include:

- How to perform common user tasks, such as searching the resource, or accessing the details of an entry
- Answers to questions that are often asked or can be anticipated, *e.g.*, with Frequently Asked Questions or tips and tricks pages
- Who to contact for questions or help

A separate section in these guidelines on the *Conditions of use policy* covers terms of use of the resource and how best to cite records in a resource and the resource itself.

Guidance for users who wish to contribute software is covered in the next section, *Provide guidance to software contributors*.

When writing guidelines for users, it is advisable to identify the types of users your resource has or could potentially have and corresponding use cases. Guidance itself should be offered in multiple forms, such as in-field prompts, linked explanations, and completed examples. Any machine-readable access, such as an API, should be fully described directly in the interface or by providing a pointer to existing documentation, and should specify which formats are supported (*e.g.*, JSON-LD, XML) through content negotiation if it is enabled.

Examples of such elements include, for instance, the bio.tools registry (*Ison et al., 2019*) API user guide (https://biotools.readthedocs.io/en/latest/api_usage_guide.html), or the ORNL DAAC (*ORNL, 2013*) instructions for data providers (https://daac.ornl.gov/submit/). Additional examples of user guidance can be found in Appendix B.

## Provide guidance to software contributors

Most software registries and repositories rely on a community model, whereby external contributors will provide software entries to the resource. The scope statement will already have explained *what* is accepted and what is not; the contributor policy addresses *who* can add or change software entries and the processes involved.

The contributor policy should therefore describe:

- Who can or cannot submit entries and/or metadata
- Required and optional metadata expected for deposited software
- Review process, if any
- Curation process, if any
- Procedures for updates (*e.g.*, who can do it, when it is done, how is it done)

Topics to consider when writing a contributor policy include whether the author(s) of a software entry will be contacted if the contributor is not also an author and whether contact is a condition or side-effect of the submission. Additionally, a contributor policy should specify how persistent identifiers are assigned (if used) and should state that depositors must comply with all applicable laws and not be intentionally malicious.

Such material is provided in resources such as the Computational Infrastructure for Geodynamics (*Hwang & Kellogg, 2017*) software contribution checklist (https://github.com/geodynamics/best_practices/blob/master/ContributingChecklist.md#contributing-software) and the CoMSES Net Computational Model Library (*Janssen et al., 2008*) model archival tutorial (https://forum.comses.net/t/archiving-your-model-1-gettingstarted/7377). Additional examples of guidance for software contributors can be found in Appendix B.

## Establish an authorship policy

Because research software is often a research product, it is important to report authorship accurately, as it allows for proper scholarly credit and other types of attributions (*Smith, Katz & Niemeyer, 2016*). However, even though authorship should be defined at the level of a given project, it can prove complicated to determine (*Alliez et al., 2019*). Roles in software development can widely vary as contributors change with time and versions, and contributions are difficult to gauge beyond the "commit," giving rise to complex situations. In this context, establishing a dedicated policy ensures that people are given due credit for their work. The policy also serves as a document that administrators can turn to in case disputes arise and allows proactive problem mitigation, rather than having to resort to reactive interpretation. Furthermore, having an authorship policy mirrors similar policies by journals and publishers and thus is part of a larger trend. Note that the authorship policy will be communicated at least partially to users through guidance provided to software contributors. Resource maintainers should ensure this policy remains consistent with the citation policies for the registry or repository (usually, the citation requirements for each piece of research software are under the authority of its owners).

The authorship policy should specify:

- How authorship is determined *e.g.*, a stated criteria by the contributors and/or the resource
- Policies around making changes to authorship
- The conflict resolution processes adopted to handle authorship disputes

When defining an authorship policy, resource maintainers should take into consideration whether those who are not coders, such as software testers or documentation maintainers, will be identified or credited as authors, as well as criteria for ordering the list of authors in cases of multiple authors, and how the resource handles large numbers of authors and group or consortium authorship. Resources may also include guidelines about how changes to authorship will be handled so each author receives proper credit for their contribution. Guidelines can help facilitate determining every contributors' role. In particular, the use of a credit vocabulary, such as the Contributor Roles Taxonomy (*Allen, O'Connell & Kiermer, 2019*), to describe authors' contributions should be considered for this purpose (http://credit.niso.org/).

An example of authorship policy is provided in the Ethics Guidelines (https://joss.theoj.org/about#ethics) and the submission guide authorship section (https://joss.readthedocs.io/en/latest/submitting.html#authorship) of the *Journal of Open Source Software* (*Katz, Niemeyer & Smith, 2018*), which provides rules for inclusion in the authors list. Additional examples of authorship policies can be found in Appendix B.

## Document and share your metadata schema

The structure and semantics of the information stored in registries and repositories is sometimes complex, which can hinder the clarity, discovery, and reuse of the entries included in these resources. Publicly posting the metadata schema used for the entries helps individual and organizational users interested in a resource's information understand the structure and properties of the deposited information. The metadata structure helps to inform users how to interact with or ingest records in the resource. A metadata schema mapped to other schemas and an API specification can improve the interoperability between registries and repositories.

This practice should specify:

- The schema used and its version number. If a standard or community schema, such as CodeMeta (*Jones et al., 2017*) or schema.org (*Guha, Brickley & Macbeth, 2016*) is used, the resource should reference its documentation or official website. If a custom schema is used, formal documentation such as a description of the schema and/or a data dictionary should be provided.
- Expected metadata when submitting software, including which fields are required and which are optional, and the format of the content in each field.

To improve the readability of the metadata schema and facilitate its translation to other standards, resources may provide a mapping (from the metadata schema used in the

resource) to published standard schemas, through the form of a "cross-walk" (*e.g.*, the CodeMeta cross-walk (https://codemeta.github.io/crosswalk/)) and include an example entry from the repository that illustrates all the fields of the metadata schema. For instance, extensive documentation (https://biotoolsschema.readthedocs.io/en/latest/) is available for the biotoolsSchema (*Ison et al., 2021*) format, which is used in the bio.tools registry. Another example is the OntoSoft vocabulary (http://ontosoft.org/software), used by the OntoSoft registry (*Gil, Ratnakar & Garijo, 2015*; *Gil et al., 2016*) and available in both machine-readable and human readable formats. Additional examples of metadata schemas can be found in Appendix B.

## Stipulate conditions of use

The *conditions of use* document the terms under which users may use the contents provided by a website. In the case of software registries and repositories, these conditions should specifically state how the metadata regarding the entities of a resource can be used, attributed, and/or cited, and provide information about the licenses used for the code and binaries. This policy can forestall potential liabilities and difficulties that may arise, such as claims of damage for misinterpretation or misapplication of metadata. In addition, the conditions of use should clearly state how the metadata can and cannot be used, including for commercial purposes and in aggregate form.

This document should include:

- Legal disclaimers about the responsibility and liability borne by the registry or repository
- License and copyright information, both for individual entries and for the registry or repository as a whole
- Conditions for the use of the metadata, including prohibitions, if any
- Preferred format for citing software entries
- Preferred format for attributing or citing the resource itself

When writing conditions of use, resource maintainers might consider what license governs the metadata, if licensing requirements apply for findings and/or derivatives of the resource, and whether there are differences in the terms and license for commercial *vs* noncommercial use. Restrictions on the use of the metadata may also be included, as well as a statement to the effect that the registry or repository makes no guarantees about completeness and is not liable for any damages that could arise from the use of the information. Technical restrictions, such as conditions of use of the API (if one is available), may also be mentioned.

Conditions of use can be found for instance for DOE CODE (*Ensor et al., 2017*), which in addition to the general conditions of use (https://www.osti.gov/disclaim) specifies that the rules for usage of the hosted code (https://www.osti.gov/doecode/faq#are-there-restrictions) are defined by their respective licenses. Additional examples of conditions of use policies can be found in Appendix B.

## State a privacy policy

Privacy policies define how personal data about users are stored, processed, exchanged or removed. Having a privacy policy demonstrates a strong commitment to the privacy of users of the registry or repository and allows the resource to comply with the legal requirement of many countries in addition to those a home institution and/or funding agencies may impose.

The privacy policy of a resource should describe:

- What information is collected and how long it is retained
- How the information, especially any personal data, is used
- Whether tracking is done, what is tracked, and how (*e.g.*, Google Analytics)
- Whether cookies are used

When writing a privacy policy, the specific personal data which are collected should be detailed, as well as the justification for their resource, and whether these data are sold and shared. Additionally, one should list explicitly the third-party tools used to collect analytic information and potentially reference their privacy policies. If users can receive emails as a result of visiting or downloading content, such potential solicitations or notifications should be announced. Measures taken to protect users' privacy and whether the resource complies with the *European Union Directive on General Data Protection Regulation* (https://gdpr-info.eu/) (GDPR) or other local laws, if applicable, should be explained[2]. As a precaution, the statement can reserve the right to make changes to this privacy policy. Finally, a mechanism by which users can request the removal of such information should be described.

For example, the SciCrunch's (*Grethe et al., 2014*) privacy policy (https://scicrunch.org/page/privacy) details what kind of personal information is collected, how it is collected, and how it may be reused, including by third-party websites through the use of cookies. Additional examples of privacy policies can be found in Appendix B.

## Provide a retention policy

Many software registries and repositories aim to facilitate the discovery and accessibility of the objects they describe, *e.g.*, enabling search and citation, by making the corresponding records permanently accessible. However, for various reasons, even in such cases maintainers and curators may have to remove records. Common examples include removing entries that are outdated, no longer meet the scope of the registry, or are found to be in violation of policies. The resource should therefore document retention goals and procedures so that users and depositors are aware of them.

The retention policy should describe:

- The length of time metadata and/or files are expected to be retained;
- Under what conditions metadata and/or files are removed;
- Who has the responsibility and ability to remove information;
- Procedures to request that metadata and/or files be removed.

[2] In the case of GDPR, the regulation applies to all European user personal data, even if the resource is not located in Europe.

The policy should take into account whether best practices for persistent identifiers are followed, including resolvability, retention, and non-reuse of those identifiers. The retention time provided by the resource should not be too prescriptive (*e.g.*, "for the next 10 years"), but rather it should fit within the context of the underlying organization(s) and its funding. This policy should also state who is allowed to edit metadata, delete records, or delete files, and how these changes are performed to preserve the broader consistency of the registry. Finally, the process by which data may be taken offline and archived as well as the process for its possible retrieval should be thoroughly documented.

As an example, Bioconductor (*Gentleman et al., 2004*) has a deprecation process through which software packages are removed if they cannot be successfully built or tested, or upon specific request from the package maintainer. Their policy (https://bioconductor.org/developers/package-end-of-life/) specifies who initiates this process and under which circumstances, as well as the successive steps that lead to the removal of the package. Additional examples of retention policies can be found in Appendix B.

## Disclose your end-of-life policy

Despite their usefulness, the long-term maintenance, sustainability, and persistence of online scientific resources remains a challenge, and published web services or databases can disappear after a few years (*Veretnik, Fink & Bourne, 2008*; *Kern, Fehlmann & Keller, 2020*). Sharing a clear end-of-life policy increases trust in the community served by a registry or repository. It demonstrates a thoughtful commitment to users by informing them that provisions for the resource have been considered should the resource close or otherwise end its services for its described artifacts. Such a policy sets expectations and provides reassurance as to how long the records within the registry will be findable and accessible in the future.

This policy should describe:

- Under what circumstances the resource might end its services;
- What consequences would result from closure;
- What will happen to the metadata and/or the software artifacts contained in the resource in the event of closure;
- If long-term preservation is expected, where metadata and/or software artifacts will be migrated for preservation;
- How a migration will be funded.

Publishing an end-of-life policy is an opportunity to consider, in the event a resource is closed, whether the records will remain available, and if so, how and for whom, and under which conditions, such as archived status or "read-only." The restrictions applicable to this policy, if any, should be considered and detailed. Establishing a formal agreement or memorandum of understanding with another registry, repository, or institution to receive and preserve the data or project, if applicable, might help to prepare for such a liability.

Examples of such policies include the Zenodo end-of-life policy (https://help.zenodo.org/), which states that if Zenodo ceases its services, the data hosted in the resource will be

migrated and the DOIs provided would be updated to resolve to the new location (currently unspecified). Additional examples of end-of-life policies can be found in Appendix B.

A summary of the practices presented in this section can be found in Table 2.

## DISCUSSION

The best practices described above serve as a guide for repositories and registries to provide better service to their users, ranging from software developers and researchers to publishers and search engines, and enable greater transparency about the operation of their described resources. Implementing our practices provides users with significant information about *how* different resources operate, while preserving important institutional knowledge, standardizing expectations, and guiding user interactions.

For instance, a public scope statement and guidance for users may directly impact usability and, thus, the popularity of the repository. Resources including tools with a simple design and unambiguous commands, as well as infographic guides or video tutorials, ease the learning curve for new users. The guidance for software contributions, conditions of use, and sharing the metadata schema used may help eager users contribute new functionality or tools, which may also help in creating a community around a resource. A privacy policy has become a requirement across geographic boundaries and legal jurisdictions. An authorship policy is critical in facilitating collaborative work among researchers and minimizing the chances for disputes. Finally, retention and end-of-life policies increase the trust and integrity of a repository service.

Policies affecting a single community or domain were deliberately omitted when developing the best practices. First, an exhaustive list would have been a barrier to adoption and not applicable to every repository since each has a different perspective, audience, and motivation that drives policy development for their organization. Second, best practices that regulate the content of a resource are typically domain-specific to the artifact and left to resources to stipulate based on their needs. Participants in the 2019 Scientific Software Registry Collaboration Workshop were surprised to find that only four metadata elements were shared by all represented resources[3]. The diversity of our resources precludes prescriptive requirements, such as requiring specific metadata for records, so these were also deliberately omitted in the proposed best practices.

Hence, we focused on broadly applicable practices considered important by various resources. For example, amongst the participating registries and repositories, very few had codes of conduct that govern the behavior of community members. Codes of conduct are warranted if resources are run as part of a community, especially if comments and reviews are solicited for deposits. In contrast, a code of conduct would be less useful for resources whose primary purpose is to make software and software metadata available for reuse. However, this does not negate their importance and their inclusion as best practices in other arenas concerning software.

As noted by the FAIR4RS movement, software is different than data, motivating the need for a separate effort to address software resources (*Lamprecht et al., 2020*; *Katz et al., 2016*). Even so, there are some similarities, and our effort complements and aligns well

[3] The elements were: software name, description, keywords, and URL.

**Table 2 Summary of the best practices with recommendations and examples.**

| Practice, description and examples | Recommendations |
|---|---|
| 1. Provide a public scope statement | • What is accepted, and acceptable, based on criteria covering scientific discipline, technical characteristics, and administrative properties |
| Informs both software depositor and resource seeker what the collection does and does not contain. | • What is not accepted, *i.e.*, characteristics that preclude their incorporation in the resource |
| Example: ASCL editorial policy. | • Notable exceptions to these rules, if any |
| 2. Provide guidance for users | • How to perform common user tasks, like searching for collection, or accessing the details of an entry |
| Helps users accessing a resource understand how to perform tasks like searching, browsing, and retrieving software entries. | • Answers to questions that are often asked or can be anticipated |
| Example: bio.tools registry API user guide. | • Point of contact for help and questions |
| 3. Provide guidance to software contributors | • Who can or cannot submit entries and/or metadata |
| Specifies who can add or change software entries and explains the necessary processes. | • Required and optional metadata expected from software contributors |
| Example: Computational Infrastructure for Geodynamics contribution checklist. | • Procedures for updates, review process, curation process |
| 4. Establish an authorship policy | • How authorship is determined *e.g.*, a stated criteria by the contributors and/or the resource |
| Ensures that contributors are given due credit for their work and to resolve disputes in case of conflict. | • Policies around making changes to authorship |
| Example: JOSS authorship policy. | • Define the conflict resolution processes |
| 5. Document and share your metadata schema | • Specify the used schema and its version number. Add reference to its documentation or official website. If a custom schema is used, provide documentation. |
| Revealing the metadata schema used helps users understand the structure and properties of the deposited information. | • Expected metadata when submitting software |
| Example: OntoSoft vocabulary from the OntoSoft registry. | |
| 6. Stipulate conditions of use | • Legal disclaimers about the responsibility and liability borne by the resource |
| Documents the terms under which users may use the provided resources, including metadata and software. | • License and copyright information, both for individual entries and for the resource as a whole |
| Example: DOE CODE acceptable use policy. | • Conditions for the use of the metadata, including prohibitions, if any |
| | • Preferred format for citing software entries; preferred format for attributing or citing the resource itself |
| 7. State a privacy policy | • What information is collected and how long it is retained |
| Defines how personal data about users are stored, processed, exchanged, or removed. | • How the information, especially any personal data, is used |
| Example: SciCrunch's privacy policy. | • Whether tracking is done, what is tracked, and how; whether cookies are used |
| 8. Provide a retention policy | • The length of time metadata and/or files are expected to be retained |
| Helps both users and depositors understand and anticipate retention goals and procedures. | • Under what conditions metadata and/or files are removed |
| Example: Bioconductor package deprecation. | • Who has the responsibility and ability to remove information; procedures to request that metadata and/or files be removed |
| 9. Disclose end-of-life policy | • Circumstances under which the resource might end its services |
| Informs both users and depositors of how long the records within the resource will be findable and accessible in the future. | • What consequences would result from closure |
| Example: Zenodo end-of-life policy. | • What will happen to the metadata and/or the software artifacts contained in the resource in the event of closure |
| | • If long-term preservation is expected, where metadata and/or software artifacts will be migrated for preservation; how a migration will be funded |

 

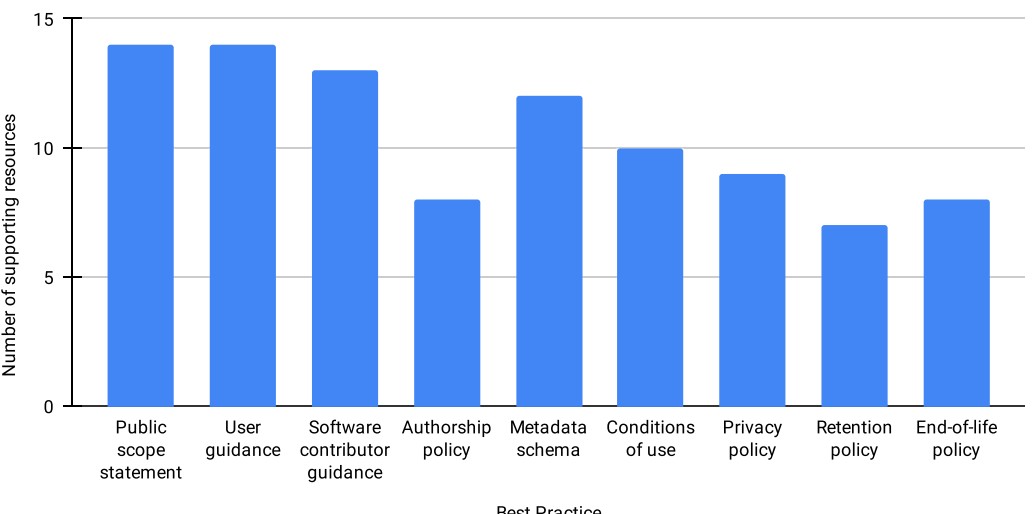

**Figure 1 Number of resources supporting each best practice, out of 14 resources.**

with recent guidelines developed in parallel to increase the transparency, responsibility, user focus, sustainability, and technology of data repositories. For example, both the TRUST Principles (*Lin et al., 2020*) and CoreTrustSeal Requirements (*CoreTrustSeal, 2019*) call for a repository to provide information on its scope and list the terms of use of its metadata to be considered compliant with TRUST or CoreTrustSeal, which aligns with our practices "*Provide a public scope statement*" and "*Stipulate conditions of use*". CoreTrustSeal and TRUST also require that a repository consider continuity of access, which we have expressed as the practice to "*Disclosing your end-of-life policy*". Our best practices differ in that they do not address, for example, staffing needs nor professional development for staff, as CoreTrustSeal requires, nor do our practices address protections against cyber or physical security threats, as the TRUST principles suggest. Inward-facing policies, such as documenting internal workflows and practices, are generally good in reducing operational risks, but internal management practices were considered out of scope of our guidelines.

Figure 1 shows the number of resources that support (partially or in their totality) each best practice. Though we see the proposed best practices as critical, many of the repositories that have actively participated in the discussions (14 resources in total) have yet to implement every one of them. We have observed that the first three practices (providing public scope statement, add guidance for users and for software contributors) have the widest adoption, while the retention, end-of-life, and authorship policy the least. Understanding the lag in the implementation across all of the best practices requires further engagement with the community.

Improving the adoption of our guidelines is one of the goals of SciCodes (http:// scicodes.net), a recent consortium of scientific software registries and repositories. SciCodes evolved from the Task Force as a permanent community to continue the dialogue and share information between domains, including sharing of tools and ideas. SciCodes

has also prioritized improving software citation (complementary to the efforts of the FORCE11 SCIWG) and tracking the impact of metadata and interoperability. In addition, SciCodes aims to understand barriers to implementing policies, ensure consistency between various best practices, and continue advocacy for software support by continuing dialogue between registries, repositories, researchers, and other stakeholders.

## CONCLUSIONS

The dissemination and preservation of research material, where repositories and registries play a key role, lies at the heart of scientific advancement. This article introduces nine best practices for research software registries and repositories. The practices are an outcome of a Task Force of the FORCE11 Software Citation Implementation Working Group and reflect the discussion, collaborative experiences, and consensus of over 30 experts and 14 resources.

The best practices are non-prescriptive, broadly applicable, and include examples and guidelines for their adoption by a community. They specify establishing the working domain (scope) and guidance for both users and software contributors, address legal concerns with privacy, use, and authorship policies, enhance usability by encouraging metadata sharing, and set expectations with retention and end-of-life policies. However, we believe additional work is needed to raise awareness and adoption across resources from different scientific disciplines. Through the SciCodes consortium, our goal is to continue implementing these practices more uniformly in our own registries and repositories and reduce the burdens of adoption. In addition to completing the adoption of these best practices, SciCodes will address topics such as tracking the impact of good metadata, improving interoperability between registries, and making our metadata discoverable by search engines and services such as Google Scholar, ORCID, and discipline indexers.

## APPENDIX A: CONTRIBUTORS

The following people contributed to the development of this article through participation in the Best Practices Task Force meetings, 2019 Scientific Software Registry Collaboration Workshop, and/or SciCodes Consortium meetings:

Alain Monteil, Inria, HAL; Software Heritage

Alejandra Gonzalez-Beltran, Science and Technology Facilities Council, UK Research and Innovation, Science and Technology Facilities Council

Alexandros Ioannidis, CERN, Zenodo

Alice Allen, University of Maryland, College Park, Astrophysics Source Code Library

Allen Lee, Arizona State University, CoMSES Net Computational Model Library

Ana Trisovic, Harvard University, DataVerse

Anita Bandrowski, UCSD, SciCrunch

Bruce E. Wilson, Oak Ridge National Laboratory, ORNL Distributed Active Archive Center for Biogeochemical Dynamics

Bryce Mecum, NCEAS, UC Santa Barbara, CodeMeta

Caifan Du, iSchool, University of Texas at Austin, CiteAs

Carly Robinson, US Department of Energy, Office of Scientific and Technical Information, DOE CODE

Daniel Garijo, Universidad Politécnica de Madrid (formerly at Information Sciences Institute, University of Southern California), Ontosoft

Daniel S. Katz, University of Illinois at Urbana-Champaign, Associate EiC for JOSS, FORCE11 Software Citation Implementation Working Group, co-chair

David Long, Brigham Young University, IEEE GRS Remote Sensing Code Library

Genevieve Milliken, NYU Bobst Library, IASGE

Hervé Ménager, Hub de Bioinformatique et Biostatistique—Département Biologie Computationnelle, Institut Pasteur, ELIXIR bio.tools

Jessica Hausman, Jet Propulsion Laboratory, PO.DAAC

Jurriaan H. Spaaks, Netherlands eScience Center, Research Software Directory

Katrina Fenlon, University of Maryland, iSchool

Kristin Vanderbilt, Environmental Data Initiative, IMCR

Lorraine Hwang, University California Davis, Computational Infrastructure for Geodynamics

Lynn Davis, US Department of Energy, Office of Scientific and Technical Information, DOE CODE

Martin Fenner, Front Matter (formerly at DataCite), FORCE11 Software Citation Implementation Working Group, co-chair

Michael R. Crusoe, CWL, Debian-Med

Michael Hucka, California Institute of Technology, SBML; COMBINE

Mingfang Wu, Australian Research Data Commons, Australian Research Data Commons

Morane Gruenpeter, Inria, Software Heritage

Moritz Schubotz, FIZ Karlsruhe - Leibniz-Institute for Information Infrastructure, swMATH

Neil Chue Hong, Software Sustainability Institute/University of Edinburgh, Software Sustainability Institute; FORCE11 Software Citation Implementation Working Group, co-chair

Pete Meyer, Harvard Medical School, SBGrid; BioGrids

Peter Teuben, University of Maryland, College Park, Astrophysics Source Code Library

Piotr Sliz, Harvard Medical School, SBGrid; BioGrids

Sara Studwell, US Department of Energy, Office of Scientific and Technical Information, DOE CODE

Shelley Stall, American Geophysical Union, AGU Data Services

Stephan Druskat, German Aerospace Center (DLR)/University Jena/Humboldt-Universität zu Berlin, Citation File Format

Ted Carnevale, Neuroscience Department, Yale University, ModelDB
Tom Morrell, Caltech Library, CaltechDATA
Tom Pollard, MIT/PhysioNet, PhysioNet

## APPENDIX B: POLICY EXAMPLES

### Scope statement

- Astrophysics Source Code Library. (n.d.). *Editorial policy*.

https://ascl.net/wordpress/submissions/editiorial-policy/

- bio.tools. (n.d.). *Curators Guide*.

https://biotools.readthedocs.io/en/latest/curators_guide.html

- Caltech Library. (2017). *Terms of Deposit*.

https://data.caltech.edu/terms

- Caltech Library. (2019). *CaltechDATA FAQ*.

https://www.library.caltech.edu/caltechdata/faq

- Computational Infrastructure for Geodynamics. (n.d.). *Code Donation*.

https://geodynamics.org/cig/dev/code-donation/

- CoMSES Net Computational Model Library. (n.d.). *Frequently Asked Questions*.

https://www.comses.net/about/faq/#model-library

- ORNL DAAC for Biogeochemical Dynamics. (n.d.). *Data Scope and Acceptance Policy*.

https://daac.ornl.gov/submit/

- RDA Registry and Research Data Australia. (2018). *Collection*. ARDC Intranet.

https://intranet.ands.org.au/display/DOC/Collection

- Remote Sensing Code Library. (n.d.). *Submit*.

https://rscl-grss.org/submit.php

- SciCrunch. (n.d.). *Curation Guide for SciCrunch Registry*.

https://scicrunch.org/page/Curation%20Guidelines

- U.S. Department of Energy: Office of Scientific and Technical Information. (n.d.-a). *DOE CODE: Software Policy*. https://www.osti.gov/doecode/policy

- U.S. Department of Energy: Office of Scientific and Technical Information. (n.d.-b). *FAQs*. OSTI.GOV.

https://www.osti.gov/faqs

### Guidance for users

- Astrophysics Source Code Library. (2021). *Q & A*

https://ascl.net/home/getwp/898

- bio.tools. (2021). *API Reference*

https://biotools.readthedocs.io/en/latest/api_reference.html

- Caltech Library. (2019). *CaltechDATA FAQ.*

https://www.library.caltech.edu/caltechdata/faq

- Harvard Dataverse. (n.d.). *Curation and Data Management Services*

https://support.dataverse.harvard.edu/curation-services

- OntoSoft. (n.d.). *An Intelligent Assistant for Software Publication*

https://ontosoft.org/users.html

- ORNL DAAC for Biogeochemical Dynamics. (n.d.). *Learning*

https://daac.ornl.gov/resources/learning/

- U.S. Department of Energy: Office of Scientific and Technical Information. (n.d.). *FAQs.* OSTI.GOV.

https://www.osti.gov/doecode/faq

## Guidance for software contributors

- Astrophysics Source Code Library. (n.d.) *Submit a code.*

https://ascl.net/code/submit

- bio.tools. (n.d.) *Quick Start Guide*

https://biotools.readthedocs.io/en/latest/quickstart_guide.html

- Computational Infrastructure for Geodynamics. *Contributing Software*

https://geodynamics.org/cig/dev/code-donation/checklist/

- CoMSES Net Computational Model Library (2019) *Archiving your model: 1. Getting Started*

https://forum.comses.net/t/archiving-your-model-1-getting-started/7377

- Harvard Dataverse. (n.d.) *For Journals.*

https://support.dataverse.harvard.edu/journals

## Authorship

- Committee on Publication Ethics: COPE. (2020a). *Authorship and contributorship.*
https://publicationethics.org/authorship

- Committee on Publication Ethics: COPE. (2020b). *Core practices.*
https://publicationethics.org/core-practices

- Dagstuhl EAS Specification Draft. (2016). *The Software Credit Ontology.*
https://dagstuhleas.github.io/SoftwareCreditRoles/doc/index-en.html#

- Journal of Open Source Software. (n.d.). *Ethics Guidelines.*
https://joss.theoj.org/about#ethics

- ORNL DAAC (n.d) *Authorship Policy*.

https://daac.ornl.gov/submit/

- PeerJ Journals. (n.d.-a). *Author Policies*.

https://peerj.com/about/policies-and-procedures/#author-policies

- PeerJ Journals. (n.d.-b). *Publication Ethics*.

https://peerj.com/about/policies-and-procedures/#publication-ethics

- PLOS ONE. (n.d.). *Authorship*.

https://journals.plos.org/plosone/s/authorship

- National Center for Data to Health. (2019). The Contributor Role Ontology.

https://github.com/data2health/contributor-role-ontology

## Metadata schema

- ANDS: Australian National Data Service. (n.d.). *Metadata*. ANDS.

https://www.ands.org.au/working-with-data/metadata

- ANDS: Australian National Data Service. (2016). *ANDS Guide: Metadata*.

https://www.ands.org.au/data/assets/pdf_file/0004/728041/Metadata-Workinglevel.pdf

- Bernal, I. (2019). *Metadata for Data Repositories*.

https://doi.org/10.5281/zenodo.3233486

- bio.tools. (2020). *Bio-tools/biotoolsSchema* [HTML].

https://github.com/bio-tools/biotoolsSchema (Original work published 2015)

- bio.tools. (2019). *BiotoolsSchema documentation*.

https://biotoolsschema.readthedocs.io/en/latest/

- The CodeMeta crosswalks. (n.d.)

https://codemeta.github.io/crosswalk/

- Citation File Format (CFF). (n.d.)

https://doi.org/10.5281/zenodo.1003149

- The DataVerse Project. (2020). DataVerse 4.0+ Metadata Crosswalk.

https://docs.google.com/spreadsheets/d/10Luzti7svVTVKTA-px27oq3RxCUM-QbiTkm8iMd5C54

- OntoSoft. (2015). *OntoSoft Ontology*.

https://ontosoft.org/ontology/software/

- Zenodo. (n.d.-a). *Schema for Depositing*.

https://zenodo.org/schemas/records/record-v1.0.0.json

- Zenodo. (n.d.-b). *Schema for Published Record*.

https://zenodo.org/schemas/deposits/records/legacyrecord.json

## Conditions of use policy

- Allen Institute. (n.d.). *Terms of Use.*
https://alleninstitute.org/legal/terms-use/
- Europeana. (n.d.). *Usage Guidelines for Metadata.* Europeana Collections.
https://www.europeana.eu/portal/en/rights/metadata.html
- U.S. Department of Energy: Office of Scientific and Technical Information. (n.d.). *DOE CODE FAQ: Are there restrictions on the use of the material in DOE CODE?*
https://www.osti.gov/doecode/faq#are-there-restrictions
- Zenodo. (n.d.). *Terms of Use.*
https://about.zenodo.org/terms/

## Privacy policy

- Allen Institute. (n.d.). *Privacy Policy.*
https://alleninstitute.org/legal/privacy-policy/
- CoMSES Net. (n.d.). *Data Privacy Policy.*
https://www.comses.net/about/data-privacy/
- Nature. (2020). *Privacy Policy.*
https://www.nature.com/info/privacy
- Research Data Australia. (n.d.). *Privacy Policy.*
https://researchdata.ands.org.au/page/privacy
- SciCrunch. (2018). *Privacy Policy.* SciCrunch.
https://scicrunch.org/page/privacy
- Science Repository. (n.d.). *Privacy Policies.*
https://www.sciencerepository.org/privacy
- Zenodo. (n.d.). *Privacy policy.*
https://about.zenodo.org/privacy-policy/

## Retention policy

- Bioconductor. (2020). *Package End of Life Policy.*
https://bioconductor.org/developers/package-end-of-life/
- Caltech Library. (n.d.). *CaltechDATA FAQ.*
https://www.library.caltech.edu/caltechdata/faq
- CoMSES Net Computational Model Library. (n.d.). *How long will models be stored in the Computational Model Library?*
https://www.comses.net/about/faq/

- Dryad. (2020). *Dryad FAQ - Publish and Preserve your Data.*
https://datadryad.org/stash/faq#preserved
- Software Heritage. (n.d.). *Content policy.*
https://www.softwareheritage.org/legal/content-policy/
- Zenodo. (n.d.). *General Policies v1.0.*
https://about.zenodo.org/policies/

### End-of-life policy

- Figshare. (n.d.). *Preservation and Continuity of Access Policy.*
https://knowledge.figshare.com/articles/item/preservation-and-continuity-of-access-policy
- Open Science Framework. (2019). *FAQs.* OSF Guides.
http://help.osf.io/hc/en-us/articles/360019737894-FAQs
- NASA Earth Science Data Preservation Content Specification (n.d.)
https://earthdata.nasa.gov/esdis/eso/standards-and-references/preservation-content-spec
- Zenodo. (n.d.). *Frequently Asked Questions.*
https://help.zenodo.org/

## APPENDIX C: RESOURCE INFORMATION

Since the first Task Force meeting was held in 2019, we have asked new resource representatives joining our community to provide the information shown in Table C.1. Thanks to this effort, the group has been able to learn about each resource, identify similarities and differences, and thus better inform our meeting discussions.

Tables C.2–C.4 provide an updated overview of the main features of all resources currently involved in the discussion and implementation of the best practices (30 resources in total as of December, 2021). Participating resources are diverse, and belong to a variety of discipline-specific (*e.g.*, neurosciences, biology, geosciences, *etc.*) and domain generic repositories. Curated resources tend to have a lower number of software entries. Most resources have been created in the last 20 years, with the oldest resource dating from 1991. Most resources accept a software deposit, support DOIs to identify their entries, are actively curated, and can be used to cite software.

**Table C.1 Questions asked to resource representatives.**

| Question | Answer type |
|---|---|
| Repository name and abbreviation | Text |
| Repository home page | URL |
| Representative name and email address | Text |
| Is the repository discipline-specific? | Yes/No |
| Is the repository for discipline software only? | Yes/No |
| Is a software deposit accepted? | Yes/No |
| Is a software deposit required? | Yes/No |
| Does your resource have a public scope/editorial policy? | URL |
| Supported unique identifier(s) type(s) | Text |
| Can the repository mint DOIs? | Yes/No |
| Is the repository actively curated? | Yes/No/Other |
| How are entries added? | Text |
| Is your resource currently used to cite software? | Yes/No/Other |
| When did your resource start operating? | Year started |
| What is the number of records (as of filling date)? | Integer |
| Notes/comments/additional information | Text |

**Table C.2 Information shared by 30 resources participating in the SciCodes consortium, as of December 2021.**

| Question | #Yes | #No | #Other |
|---|---|---|---|
| Is the resource discipline-specific? | 18 | 12 | 0 |
| Does the resource accept software only? | 17 | 13 | 0 |
| Does the resource require a software deposit? | 5 | 25 | 0 |
| Does the resource accept a software deposit | 22 | 8 | 0 |
| Can the resource mint DOIs? | 16 | 14 | 0 |
| Is the resource actively curated? | 21 | 3 | 6 |
| Can the resource be used to cite software? | 21 | 6 | 3 |

**Table C.3 Number of entries described in the resources of the SciCodes consortium, by December 2021.**

| #Entries | #Resources |
|---|---|
| Unknown | 2 |
| Less than 1K | 15 |
| 1K–100K | 10 |
| More than 100K | 3 |

Table C.4 **Date of creation of the resources in the SciCodes consortium, by December 2021.**

| Creation year | #Resources |
|---|---|
| Before 2000 | 6 |
| 2000–2010 | 9 |
| After 2010 | 15 |

## ACKNOWLEDGEMENTS

The best practices presented here were proposed and developed by a Task Force of the FORCE11 Software Citation Implementation Working Group. The following authors, randomly ordered, contributed equally to discussion, conceptualization, writing, reviewing, and editing this article: Daniel Garijo, Lorraine Hwang, Hervé Ménager, Alice Allen, Michael Hucka, Thomas Morrell, and Ana Trisovic.

**Task Force on Best Practices for Software Registries participants**: Alain Monteil, Alejandra Gonzalez-Beltran, Alexandros Ioannidis, Alice Allen, Allen Lee, Andre Jackson, Bryce Mecum,Caifan Du, Carly Robinson, Daniel Garijo, Daniel Katz, Genevieve Milliken, Hervé Ménager, Jurriaan Spaaks, Katrina Fenlon, Kristin Vanderbilt, Lorraine Hwang, Michael Hucka, Neil Chue Hong, P. Wesley Ryan, Peter Teuben, Shelley Stall, Stephan Druskat, Ted Carnevale, Thomas Morrell.

**SciCodes Consortium participants**: Alain Monteil, Alejandra Gonzalez-Beltran, Alexandros Ioannidis, Alice Allen, Allen Lee, Ana Trisovic, Anita Bandrowski, Bruce Wilson, Bryce Mecum, Carly Robinson, Celine Sarr, Colin Smith, Daniel Garijo, David Long, Harry Bhadeshia, Hervé Mé nager, Jeanette M. Sperhac, Joy Ku, Jurriaan Spaaks, Kristin Vanderbilt, Lorraine Hwang, Matt Jones, Mercé Crosas, Michael R. Crusoe, Mike Hucka, Ming Fang Wu, Morane Gruenpeter, Moritz Schubotz, Olaf Teschke, Pete Meyer, Peter Teuben, Piotr Sliz, Sara Studwell, Shelley Stall, Ted Carnevale, Tom Morrell, Tom Pollard, Wolfram Sperber.

### Funding

This work was supported by the Alfred P. Sloan Foundation (Grant Number G-2019-12446), and the Heidelberg Institute of Theoretical Studies. Ana Trisovic is funded by the Alfred P. Sloan Foundation (Grant Number P-2020-13988). The funders had no role in study design, data collection and analysis, decision to publish, or preparation of the manuscript.

### Grant Disclosures

The following grant information was disclosed by the authors:
Alfred P. Sloan Foundation: G-2019-12446 and P-2020-13988.
Heidelberg Institute of Theoretical Studies.

## Competing Interests

The authors declare that they have no competing interests.

## Author Contributions

- Daniel Garijo conceived and designed the experiments, prepared figures and/or tables, authored or reviewed drafts of the article, and approved the final draft.
- Hervé Ménager conceived and designed the experiments, prepared figures and/or tables, authored or reviewed drafts of the article, and approved the final draft.
- Lorraine Hwang conceived and designed the experiments, prepared figures and/or tables, authored or reviewed drafts of the article, and approved the final draft.
- Ana Trisovic conceived and designed the experiments, prepared figures and/or tables, authored or reviewed drafts of the article, and approved the final draft.
- Michael Hucka conceived and designed the experiments, prepared figures and/or tables, authored or reviewed drafts of the article, and approved the final draft.
- Thomas Morrell conceived and designed the experiments, prepared figures and/or tables, authored or reviewed drafts of the article, and approved the final draft.
- Alice Allen conceived and designed the experiments, prepared figures and/or tables, authored or reviewed drafts of the article, and approved the final draft.

## Data Availability

There is no data or code associated with this publication.

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
