# Peer review of "Nine best practices for research software registries and repositories"

_PeerJ Computer Science, doi:10.7717/peerj-cs.1023_

## Round 0.1 · original submission · Major Revisions

Based on the opinion of two reviewers, and my own assessment as editor, I feel that your paper presents a great contribution and has good potential. You are requested to address the reviewers' comments and then your manuscript can be considered further.

Reviewer 1 ·

Basic reporting

The paper is generally well-written and easy to follow.

In terms of literature references, some relevant software archives are not cited, for example, swMATH at https://swmath.org/ , main reference:

Greuel, Gert-Martin, and Wolfram Sperber. "swMATH–an information service for mathematical software." International Congress on Mathematical Software. Springer, Berlin, Heidelberg, 2014

and some other archives are cited but only with URLs rather than also with the relevant literature references, e.g, Software Heritage is only cited via URL instead of referencing:

Di Cosmo, Roberto, and Stefano Zacchiroli. "Software heritage: Why and how to preserve software source code." iPRES 2017-14th International Conference on Digital Preservation. 2017

similarly, figshare is only mentioned with URL instead of also referencing, among others:

Thelwall, Mike, and Kayvan Kousha. "Figshare: a universal repository for academic resource sharing?." Online Information Review (2016).

Both swMATH and Software Heritage should also be mentioned in the introduction, together with Zenodo (that does have a proper citation) and FigShare.

In the introduction, I recommend to anticipate the introduction of the terminology of "resources and collections", because it is really tiresome to mention "registries and repositories", up to that point in the paper. Also, the authors should probably decide between "resources" and "collections" and be consistent about that.

In section 2, Background, "RDA" and "FORCE11" needs references, as they might not be familiar to all readers.
When mentioning "tools have been developed to facilitate depositing research data", other platforms should be mentioned, such as Zenodo, Software Heritage, and other services like the HAL preprint service that now supports depositing source code as well.
At the end of the section, I find it incorrect to say that you "address the needs of domain software registries and repositories", because you are just providing guidelines. Maybe you can tone down this claim a little.

Regarding the presentation of the best practice, it is hard to cross-reference the concrete example of each best practice to the end in the appendix.
I recommend including at least 1-2 examples *inline* in the main paper text just after each best practice, and cross-reference the appendix for *other* examples.
That way the main text of the paper become self-contained (and more interesting!) and the appendix can be consulted only for those readers who want more.
(This is a comment critique/suggestion that applies to all best practices, as they all have examples; which is definitely a good thing!)

As a minor point, the author list is inconsistent between the submission system and the paper; the latter also includes "Task Force on Best Practices for Software Registries, and SciCodes Consortium".
As these are not proper authors, they should not be listed as such, but rather included in the acknowledgments.

Experimental design

Section 3 is a bit disappointing, in the sense that it reads like poorly presented results of a survey.
You should document what were *all* the questions asked, not just a few of them ("questions [...] included") and detail what answers you got in the usual way (descriptive statistics, etc.), otherwise it would be hard to assess the pertinence of the methodology followed to arrive at the guidelines, and this would appear to be just a position paper by the authors.

Validity of the findings

About the specific best practices:

4.1: the wording is weird, the main point is the first one ("What is accepted"), the other two points feel just redundant restatement of the same notion ("what is not accepted" -> complement of the first point; and "notable exceptions" -> which is still a part of the notion of "what is accepted").
Maybe you should recommend that resources operators just focus on the properties/criteria of the artifacts that are acceptable, rather than restating the same notion in different ways.

4.3: the wording of "required and optional metadata" is weird, because it centers on the fact that metadata comes "from software contributors".
Shouldn't this be worded as "required and optional metadata expected for deposited software"?
Metadata are about the software, and the operators should care that they exist, no matter who contribute them.

4.4 "Also, particular care should be taken to maintain the consistency of this policy with the citation policies for the registry or repository." -> I have no idea what this means, it should be better explained/clarified in the text.

4.5 "share your metadata schema" -> "share" should be "publish" or "document", as it's really about making it public, not sharing with others.
"This practice when implemented should specify:" -> drop "when implemented", which feels like a truism that could be said for any of the best practices.

4.6 this best practice is almost entirely worded about metadata, and that seems incorrect to me.
The conditions of use are relevant for both metadata and the data itself (e.g., actual software, for software repositories).
The text of this should be generalized to cover both scenarii on equal footing, maybe adopting the syntactic convention of "(meta)data" when talking about both, as the FAIR Principles do.

4.7 include passages like (emphasis mine) "Additionally, one *can* list explicitly the third-party tools" and "a mechanism by which users can request the removal of such information *may* be described".
In a guideline document, I have a hard time understanding what those two modal verbs mean.
Should the resource operators include those information or not?
I recommend that the authors take a stance on this point.
(Or else describe the precise semantics of the various modal verbs use, e.g., in the style of RFC 2119.)
All this would make the guidelines much more actionable for resource operators.

4.8 When discussing taking offline data and archival, I consider there is an important omission: the requirement of documenting the backup strategy (which commonly goes under the notion of "retention policy") and how the archive reacts to legal takedown notices (e.g., due to DMCA in the US or, in Europe, equivalent legislation as well as GDPR).

4.9 When mentioning the Zenodo example, it would be nice to mention in the paper to where data will be migrated, in addition to the fact that they will be migrated.

Reviewer 2 ·

Basic reporting

no comment

Experimental design

no comment

Validity of the findings

no comment

Additional comments

I really appreciate the efforts by the Task Force of FORCE11 SCIWG to come up with the nine best practices for researching software registries and repositories, as well as the clear, well-structured report presented here that provides sufficient context to justify its significance. I have only a few minor comments listed below.

- In lines 151–155, the authors mention that the Task Force gathered information from its members to learn more about each resource and identify overlapping practices. I would encourage the authors to provide a simple visualization of their survey results as the basis for developing best practices.
- Similar to the previous point, the authors mention in lines 483–486 that they observe different adoption rates for best practices upon internal discussion. It would be great if the authors could provide some simple statistics (preferably with visualizations) to let the reader know the status quo.
- Lastly, since the authors mention in lines 487–490 that their effort complements and aligns well with recent guidelines developed for data repositories, it would be interesting to learn the actual similarities and differences between best practices developed by the two communities.

Overall, the nine best practices proposed here are quite comprehensive and tackle many important issues within the broader research community. I look forward to future impact tracking and assessments undertaken by SciCodes.

---

## Round 0.2 · Major Revisions

Thanks for the efforts in revising your manuscript. However, based on the reviewer's comments and my own assessment as editor, I recommend you incorporate the reviewer's feedback in a new revised version.

Reviewer 2 ·

Basic reporting

no comment

Experimental design

no comment

Validity of the findings

no comment

Additional comments

I would like to first commend the authors' efforts in addressing the reviewers' questions in detail and revising their manuscript accordingly. While I'm overall satisfied with the revision, if I had to be picky, I would suggest the authors integrate what is currently in Appendix C into the Methodology section. Both Reviewer #1 and I urged the authors to provide descriptive statistics and/or visualization of their initial survey results to substantiate the empirical grounds of this paper, but the rewritten Methodology section primarily supplies procedural details rather than survey details we have requested.

For instance, lines 154--156 can be re-written as: Participants introduced themselves and their resources by providing some basic information, including repository name, starting year, number of records, target audience (discipline-specific or general), as well as services provided (e.g., support of software citation, software deposits, and doi minting). Figure/Table 1 presents an overview of the responses.

Question | Yes | No | Other
--- | --- | --- | ---
Q1 | raw count (%) | raw count (%) | raw count (%)
Q2 | raw count (%) | raw count (%) | raw count (%)

You can then briefly comment on the results and direct readers to Appendix C for the full list of questions and other summaries of the results if available (e.g., distribution of supported unique identifier types, number of records, and years when the repository started operating).

On a second thought, it seems more appropriate to use a table instead of a figure to present the results, which saves space, but the final decision should be at the authors' discretion. If a figure is still preferred, I would encourage the authors to experiment with other types of visualization such as a stacked bar chart or a divergent stacked bar chart. The rationale is that we know the responses for each question sum to 30 anyway, and there are only three possible answers for each (Yes/No/Other), so there is no real benefit of having separate bars to represent possible answers for each question.

Other minor points

- The quotation marks in lines 1018--1019 do not look right, possibly an issue related to LaTex.

---

## Round 0.3 · Major Revisions

After careful screening of 2 reviewer comments, I feel that we are moving closer to destination. However, I feel that the comments of reviewers are major in nature. So I suggest your team to make a careful revision and upload revision as soon as possible. We will try to speed up the final revisions.

Reviewer 1 ·

Basic reporting

As a general comment for the authors and editors, there seems to be a process failure here. I have reviewed v0 of this paper and I'm not reviewing v2. There seem to have been a v1 in between, which I have not reviewed. As a consequence of that I have seen neither the tracked changes between v0 and v1, nor the rebuttal for v1 (which presumably included author answers to my initial remarks). Hence in this review I'm solely pointing out which parts of my initial review for v0 are still not addressed. This is not due to a fault of the authors, but of the review process. Still, as a reviewer, I have to insist on the points for which I have received neither an answer nor a change.
* * *
Regarding basic reporting, the only remaining unaddressed point is this:

> Regarding the presentation of the best practice, it is hard to cross-reference the concrete example of each best practice to the end in the appendix. I recommend including at least 1-2 examples *inline* in the main paper text just after each best practice, and cross-reference the appendix for *other* examples. That way the main text of the paper become self-contained (and more interesting!) and the appendix can be consulted only for those readers who want more. (This is a comment critique/suggestion that applies to all best practices, as they all have examples; which is definitely a good thing!)

Experimental design

No remaining unaddressed points from my initial review remains about experimental design. It's all good!

Validity of the findings

The following points of my initial review for v0 remains unaddressed (and are in fact the main reason why I am recommending a major revision):

> 4.1: the wording is weird, the main point is the first one ("What is accepted"), the other two points feel just redundant restatement of the same notion ("what is not accepted" -> complement of the first point; and "notable exceptions" -> which is still a part of the notion of "what is accepted"). Maybe you should recommend that resources operators just focus on the properties/criteria of the artifacts that are acceptable, rather than restating the same notion in different ways.

> 4.4 "Also, particular care should be taken to maintain the consistency of this policy with the citation policies for the registry or repository." -> I have no idea what this means, it should be better explained/clarified in the text.

> 4.5 "share your metadata schema" -> "share" should be "publish" or "document", as it's really about making it public, not sharing with others.

> 4.6 this best practice is almost entirely worded about metadata, and that seems incorrect to me. The conditions of use are relevant for both metadata and the data itself (e.g., actual software, for software repositories). The text of this should be generalized to cover both scenarii on equal footing, maybe adopting the syntactic convention of "(meta)data" when talking about both, as the FAIR Principles do.

> 4.8 When discussing taking offline data and archival, I consider there is an important omission: the requirement of documenting the backup strategy (which commonly goes under the notion of "retention policy") and how the archive reacts to legal takedown notices (e.g., due to DMCA in the US or, in Europe, equivalent legislation as well as GDPR).

Additional comments

Other than the above points, the authors did a good job at improving the paper. Thanks a lot for addressing all the (other) points I had raised in my initial review. I'm looking forward to this article finalization.

Reviewer 2 ·

Basic reporting

no comment

Experimental design

no comment

Validity of the findings

no comment

Additional comments

I noticed that the time of the inaugural Task Force meeting (February 2019) is different from the date shown in the Table 1 caption (November 2019). I was wondering if this was a random mistake or if the responses were actually collected later. Other than that, all the previous comments have been adequately addressed, and the manuscript can be accepted in its current form.

---

## Round 0.4 · accepted · Accept

We appreciate the efforts made by your team in addressing the reviewer comments and improving the paper. I am happy to accept your research article.